# LOGICJITTER: LET LLMs PLAY LOGIC GAMES AND THEY WILL DETECT MISINFORMATION

## ABSTRACT

In the face of the growing challenge of information overload online, the ability to accurately distinguish between genuine information and misinformation has become increasingly critical both from an individual and from a societal point of view. Methodologies for misinformation detection predominantly rely on supervised approaches, which depend heavily on large labeled datasets. However, these datasets are not only costly and time-consuming to produce, but they are also susceptible to issues such as labeling bias, time leakage, the inherent subjectivity of the task, and domain-specific limitations. In this paper, we aim to overcome the aforementioned challenges by proposing a novel and cost-effective strategy to enhance the logical reasoning capabilities of Large Language Models (LLMs), thereby improving their ability to detect misinformation. Our approach, termed LogicJitter, employs a data augmentation technique during fine-tuning that generates both correct and incorrect statements within rule-based logic games. These games are designed to counteract well-known human cognitive biases and logical fallacies. Hence, the primary contributions of this work include demonstrating the effectiveness of logical reasoning fine-tuning on LLMs and providing an open source package for the automatic generation of correct and incorrect logic-based training data, to ease reproducibility. Experimental results confirm this approach improves misinformation detection.

## 1 INTRODUCTION

In an increasingly connected and automated world, where information is generated and disseminated online at an unprecedented speed and volume, often without reliable oversight, the risk of *information manipulation* grows daily (Wardle & Derakhshan, 2017; Wu et al., 2019). As a result, distinguishing between authentic information and misinformation has become more crucial than ever. Indeed, falling into misinformation can lead to serious problems both for individuals and for society as a whole. For example, individuals may make harmful decisions about their health by following false medical advice. On a broader scale, misinformation can fuel distrust in public institutions, spread false narratives, and even incite social unrest, destabilizing communities (OECD, 2024).

Most of the approaches used in recent years in the literature for misinformation detection rely on *supervised learning* solutions (i.e., classifying information as true or false) (Hu et al., 2022b; Reis et al., 2019; Viviani & Pasi, 2017). However, these approaches come with several challenges. First, they require large amounts of labeled data, which is often costly and time-consuming to obtain (Reis et al., 2019). Second, they struggle to adapt to new types of misinformation, as the specific datasets they are trained on may be domain-dependent and not generalize well to other domains (Clarke et al., 2020; D'Ulizia et al., 2021). Lastly, these models can oversimplify the issue, as misinformation is not always clearly true or false but may exist in gray areas of partial truths or misleading contexts (Cabitza et al., 2022).

Recently, there has been growing interest in using *Large Language Models* (LLMs) to address the problem of misinformation detection (Hu et al., 2024; Papageorgiou et al., 2024). However, several open challenges remain. One key issue is that LLMs can sometimes generate or amplify misinformation themselves, as they rely on vast amounts of data from various sources, including unreliable ones, possibly leading to *factual inconsidency* and *hallucinations* (Ji et al., 2023). Furthermore,

LLMs can also be prone to biases present in their training data, which can affect the accuracy and fairness of their outputs. Lastly, there are concerns about the *interpretability* and *transparency* of these models (Zhao et al., 2024), making it difficult to understand how they reach their conclusions.

It is in this context that this article proposes a novel approach to circumvent the scarcity of labeled data for misinformation detection with LLMs. We hypothesize that enhancing the *logical reasoning* abilities of an LLM can significantly improve its capability to detect misinformation. In fact, misinformation often exploits human *cognitive biases* and *fallacies* (French et al., 2023; Stanovich, 2003), which we hypothesize could be more easily identified if an LLM is trained to focus on *logical consistency* (Dillehay et al., 1966; Kainz, 1995). Therefore, we propose and evaluate the fine-tuning of LLMs using carefully designed datasets that encourage logical reasoning, while simultaneously fine-tuning them for misinformation detection. Since purely natural language datasets are often ambiguous and domain-dependent, we propose training the models on *logic games* that are algorithmically generated but can be readily explained in natural language. In this logic games, proofs are presented either with or without errors, and with distractors that model situations where cognitive biases and fallacies would come into play. The LLM is then trained to detect flaws in the reasoning. The objective is to strengthen the model's logical rigor, thereby enhancing its ability to identify common human biases and, consequently, misinformation. We therefore focus on addressing the following two research questions:

> **RQ1.** Can LLM's ability to reason logically in texts be improved with rule-based logic games?
>
> **RQ2.** Will LLMs trained to detect cognitive biases and logical fallacies help in addressing the misinformation detection task?

Our proposed approach, namely *LogicJitter*, is a form of data augmentation for textual datasets, named in analogy to *ColorJitter* (Zini et al., 2023), a data augmentation technique for image datasets. In LogicJitter, true and false statements based on logic games are interspersed within the training batches to introduce variability and improve the model's reasoning capabilities.

Building on the previous premises, our key contributions are as follows:

- We demonstrate that fine-tuning LLMs to identify logical errors in structured logic games, a method we term LogicJitter, effectively enhances their ability to discern between accurate information and misinformation;
- To ease reproducibility, we provide both a PyTorch and a HuggingFace package that automatically generates correct and incorrect logic games, with the flexibility to be extended with additional logic games.

## 2 RELATED WORK

**Training LLMs to solve Rule-Based Problems.** The use of synthetic data generation has been investigated as a means of providing more data in language modeling and reasoning tasks. For instance, (Gunasekar et al., 2023) leverage LLM-generated data with notable performance on reasoning-based tasks. However, when LLMs are trained on rule-based problems, they are usually tested on the same rule-based problems. Generating synthetic geometric data for pre-training and fine-tuning the model on auxiliary constructions to address specific theorems was used to tackle Olympiad-level math problems (Trinh et al., 2024). However, LLMs seem to generalize worse in out-of-distribution data, assessed by length generalization, compared to graph networks trained on the same algorithmically generated problems (Veličković et al., 2022; Markeeva et al., 2024). Formal languages have been used to understand if LLM truly were equally capable of learning languages that are possible and impossible for humans to learn (Kallini et al., 2024). It would be however valuable to be able to leverage a rule-based training, to generalize on natural language datasets.

**LLMs at Causal and Logical Reasoning.** LLMs have been shown to be able to learn to detect structures in complex causal networks, when trained on simple ones (Vashishtha et al., 2024). Complex self learning loops have can allow LLMs to improve their ability to reason by coming up with their own rationales (Zelikman et al., 2022). Despite their progress, LLMs still struggle with complex mathematics and external computation libraries and symbolic solvers are becoming standard to

produce LLMs that are stronger in maths (Gou et al., 2024). Formal benchmarks exist to assess the causal reasoning abilities of LLMs (Jin et al., 2023; 2024), focusing on the identification of direct and indirect causal relationships. Their findings underscore the shortcomings of LLMs in executing accurate causal reasoning.

**Cognitive Biases and Fallacies.** Datasets and tasks to detect logical fallacies are available, but usually rely on human annotators that have to classify the errors encounter as a type of fallacy or as another (Jin et al., 2022). On the other hand, AI systems have been observed to reproduce some human cognitive biases such as the confirmation bias, the primacy effect, the representativeness bias, the anchoring bias, and problems related with causality (Martínez et al., 2022). Contrary to recent studies showcasing advanced reasoning abilities in LLMs, randomized controlled trials demonstrate the existence of anchoring bias in all models tested (Nguyen, 2024).

**LLMs and Misinformation Detection.** It has been shown that capabilities of LLMs to mislead humans are already superior to those of humans to mislead humans, and that the best detectors of LLM lies are LLMs themselves (Chen & Shu, 2024). However, developing datasets for fact checking is often costly given that it can require experts in the field of interest (Kotonya & Toni, 2020a). Therefore, datasets for misinformation detection are not only rare but also small (Schlichtkrull et al., 2024).

## 3  METHODOLOGY

In this section we define the cognitive biases and fallacies we are interested in compensating for (under the hypothesis that helping LLMs detect them will help them detect misinformation), we give the details of LogicJitter and the logic games we use in this work, we detail the considered LLMs, and we discuss the fine-tuning procedure.

### 3.1  COGNITIVE BIASES

Human cognitive biases are psychological tendencies that often lead to unconsciously distorted thinking and decision-making. They can be generally regrouped into (Van Eyghen, 2022; Gigerenzer, 2002): $(i)$ *belief, decision-making and behavioral*, $(ii)$ *social*, and $(iii)$ *memory* biases. Of the long list of well-studied biases, we describe those we try to compensate for with *LogicJitter*.

**Belief, decision-making, and behavioral biases.** Many biases relate to the unexpected speed and direction of updates of beliefs in human decisions. In fact, it has been shown that humans update their beliefs slower than Bayes rule, possibly a consequence of having to deal with noise in the memory recall and in the evidence acquisition processes (Hilbert, 2012). Therefore a slow belief update could be Bayes optimal in the presence of memory noise or mistrust in the sources for example, which could also be modeled as noise in the evidence acquisition process. Some biases emphasize belief updates in long-term memories, that could be stored at the synapse level, and some in short-term memories, possibly stored as spiking activity. The cognitive biases we think can better be compensated algorithmically are the tendency to revise beliefs insufficiently when presented with new evidence (*conservatism bias*), the tendency to reject evidence that contradicts established norms (*Semmelweis reflex*), to ignore the general prevalence, in favor of the information pertaining only to a specific case (*base rate fallacy*), to expect a member of a group to have certain characteristics without having actual information about that individual (*stereotyping*), to misinterpret statistical experiments involving conditional probabilities (*Berkson's paradox*) and the tendency to fail to recognize that a plan of action is no longer appropriate for a changing situation (*plan continuation bias*).

**Social biases.** Our thinking about other people often follows distorted patterns, leading us to overestimate, underestimate, or misjudge them. Some examples are the tendency to trust more the opinion of an authority figure, regardless of the content (*authority bias*), for people to seem more attractive in a group than alone (*cheerleading effect*), for someone's positive or negative traits to influence how others perceive them in unrelated areas (*halo effect*), the tendency to believe that physically attractive individuals also have intelligence, good judgment, or other positive personality traits (*physical attractiveness*) or the tendency to do and believe as others (*bandwagon effect*).

**Memory biases.** Some cognitive biases can enhance or hinder memory recall or can distort the content of the recalled memory. Some examples that we tackle by LogicJitter are the tendency to prefer easily available examples (*availability bias*), the fact that unusual or strange information is remembered more effectively than ordinary information (*bizarreness effect*) or the tendency to recall better items that are at the beginning or at the end of a list (*primacy and recency effects*).

## 3.2 FALLACIES

Fallacies are errors in logical reasoning that affect the validity of arguments, which can be either intentional or unintentional. While both cognitive biases and fallacies can lead to incorrect conclusions, cognitive biases are more about how we think, and fallacies are about how we argue.

**Formal fallacies.** They are errors in the argument's logical form in a formal logical system. One example is *illicit commutativity* which takes the form: 'If A then B. Therefore, if B then A'. Another example is *denying the antecedent*, such as: 'If A then B. Therefore, if not A then not B'. Neither of the two is generally true, which is why they are considered fallacies;

**Informal fallacies.** They are false because they rely on false premises. An example of an informal fallacy that we saw above as a cognitive bias is the *authority bias*, where an argument is deemed true only on the premise of the position of the person making the statement.

## 3.3 LOGICJITTER

To account for the data scarcity in misinformation detection and to compensate for the aforementioned forms of cognitive biases and fallacies, we introduce a data augmentation technique that we name *LogicJitter*, based on *logic games*, described textually, and *distractors* that make reasoning about them more difficult, but still possible in an exact manner.

### 3.3.1 LOGIC GAMES

We generated algorithmically the following games that are used in the fine-tuning phase to increase the reasoning ability of LLMs.

**Guided Maths.** This is a dataset with step-by-step solutions to mathematical equations. We use the *scratchpad* technique (Nye et al., 2021; Zelikman et al., 2022) to show the steps necessary to solve three types of sub-problems: *addition*, *multiplication*, and *polynomial evaluation*. Whenever we introduce an *error*, we make sure the mathematical proof we provide in the scratchpad assumes the mistake was correct and builds on top of it.

**Causal Clauses.** A dataset where we generate complex graphs of causal links, and ask if a statement is true within that graph. Each graph has between three and six nodes, one-tenth of the time the net will be linear, and the rest it will be equally likely Erdos-Renyi, Watts-Strogatz or Barabasi-Albert (Albert & Barabási, 2002), with randomized edge direction. The two subproblems we propose are: $(i)$ determining if two random nodes are connected or not, and $(ii)$ determining if a random node is a fork, a collider, none, or both.

**Context-Free Grammars.** We build random context-free grammars (Hoperoft & Ullman, 1979), that are not recurrent and have a maximum of 5 non-terminals and 4 terminals. Then we design two sub-problems: $(i)$ provided a generation, the LLM is asked if it belongs to the grammar, and $(ii)$ provided with up to 4 grammars, the LLM is asked which one can produce the most or the least amount of sentences.

**CLEVR and CLEAR.** We leverage datasets like CLEVR (Johnson et al., 2017), for visual reasoning and CLEAR (Abdelnour et al., 2018; 2023), for acoustic question-answering, to provide varied logical challenges, in this case spatial and temporal logical challenges. We use up to 8 templates, as sub-problems, to generate a diverse set of complex questions about the scenes.

### 3.3.2 DISTRACTORS

We previously introduced the first two distractors, i.e., the *diversity of games* and related *sub-problems*, and the *diversity of errors* we introduced in half of the generations.

**Errors.**     Usually, datasets algorithmically generated are created without mistakes, and human-generated textual datasets are likely to have errors but it is difficult to know if this is the case. Here we introduce mistakes algorithmically, to always know where they are. We do it because we want to train LLMs to spot subtle mistakes by themselves, and therefore in half of the samples there is going to be an error.

**Random Characters.** After presenting the problem, a solution is stated, which may be either correct or incorrect if we introduce an error. Following this, a set of randomly generated characters provide their opinions. We use up to five characters, with an equal probability of any number of them being right or wrong. Consequently, the likelihood of one character being wrong is the same as that of two being wrong, or all being wrong, with all possible outcomes equally likely.

To have a list of characters is thought to provide an opportunity to compensate for the *primacy* and *recency biases*, since the characters providing the correct answer are placed randomly. Also, all characters can be wrong, so the available information might be wrong, providing a chance to compensate for the *availability bias*.

Characters are generated in the form 'one adjective + one noun'. The adjective is picked randomly to describe either a nationality (e.g., Namibian), a similarity description (e.g., like you), a sexual orientation (e.g., bisexual), a religious affiliation (e.g., Buddhist), an ethnic group (e.g., Pacific Islander), a degree of attractiveness (e.g., good-looking), or a character trait (e.g., disrespectful). The noun is picked randomly to describe a family relationship (e.g., cousin), an authority figure (e.g., ambassador), a generic person (e.g., individual), a political orientation (e.g., libertarian), or a group (e.g., alliance). This trick gives a chance to compensate for *stereotyping*, but also for the *bizarreness effect*, given that it encourages a disassociation between personal description and being logically right or wrong. For example, sampling randomly different degrees of attractiveness compensates for the *physical attractiveness bias*, and sampling nouns for groups is intended to compensate for the *bandwagon effect*. Therefore with the random characters we intend to compensate essentially for all the social and memory biases we introduced in Section 3.1.

After presenting the problem and the opinions of the characters, the LLM is asked to provide if one of them was right or wrong, and this time the correct answer is given without error.

**Problem Revision.** After that, a modification in the initial problem statement is provided, such as new connections in the causal net, or objects removed from the CLEVR scene. The same characters appear again, to provide an opinion on the problem, and they are assigned again randomly a correct or an incorrect answer, independently from the first round. The revision is designed to compensate for the belief biases, and make the LLM take into account new evidence to revise beliefs against the *conservativism bias*, or changing the truths that condition the replies against *Berkson's paradox*, for example. Also revising the initial problem statement is a 'not A' statement, A being the initial problem, therefore providing by default 'not B' as an answer would be incorrect, which is designed to compensate for the *denying the antecedent* fallacy. We did not target directly the *illicit commutativity*, given that logically, the statement 'if B then A' is true if and only if 'not A then not B' is true, and we assume compensating *denying the antecedent* will automatically compensate for *illicit commutativity*.

The convenience of this approach is, among other things, that it should not be subject to labeling bias, time leakage, the inherent subjectivity of the task nor domain-specific limitations. For example in the case of time leakage, LogicJitter is not dependent on fine-tuning information that is potentially in the future of the misinformation that we are trying to classify. For those focused on explainability, it would be relatively straightforward to generate an explanation for the occurrence of an error within a logic game. However, given the already extensive nature of our problem descriptions, we opted not to incorporate the explainability component to avoid further complexity.

Examples of the logic games and distractors contained in LogicJitter are illustrated in Table 1. In green are highlighted the random characters that compensate for social and memory biases, that could be used stereotypically by LLMs. Given that being right or wrong is assigned randomly to the characters in the games, our goal is to train the model not to use that information to evaluate their answer, and only evaluate them within the context of the game. In blue, is highlighted the problem revision, designed to modify the content of the initial problem, and ask the LLM to reevaluate the new answers of the characters, and to reevaluate its own understanding of the scene, to compensate for belief biases and fallacies.

Table 1: LogicJitter presents textual logic games with errors and distractors, but the truth value remains exact. We highlight in green the random characters that compensate for social and memory biases, that could be used stereotypically by LLMs. We highlight in blue the problem revision, designed to compensate for belief biases and fallacies.

---

**Logic Games**

---

**Guided maths**
Input: $7x^3$ for $x = 7$ Target: $< scratch >$, $7x^3 = 7 * (7)^3 = (7) * (343) = 770$, 770, $< /scratch >$, 770. A quaint crew says it's fine. Is the quaint crew correct? False. At a second try it is shown that Input: $7x^3$ for $x = 7$ Target: $< scratch > 7x^3 = 7 * (7)^3 = (7) * (343) = 2401$, 2401, $< /scratch >$, 2365. A quaint crew says it's not ok. Is the quaint crew correct? True.

---

**Causal Clauses**
Visualize that A fixes B, B fixes C, D fixes A, D fixes C. For this reason, C fixes A. A clique from your country says it's correct, a woman from your region says it's wrong, a socialist from another region says there's no error, a queer club says it's fine, a queer crew says it's not good. Is the woman from your region correct? True. It was later brought to the attention that A does not fix B. Hence C fixes A. A socialist from another region says C doesn't fix A, a woman from your region says it's not correct, a queer crew says it's not ok, a clique from your country says it's right, a queer club says C doesn't fix A. Is the queer crew correct? True.

---

**Context-Free Grammars**
Given grammar 0, [...], grammar 1, [...], grammar 2, [...] Which grammar produces the largest number of sentences? Grammar 2. A heterosexual provost says it's not correct, a native american liberal says it's not correct, a pansexual community says grammar 1. Is the pansexual community correct? False. Grammar 0 was changed for [...] Which grammar produces the smallest number of sentences? Grammar 1. A pansexual community says it's not correct, a native american liberal says it's not correct, a heterosexual provost says there is an error. Is the heterosexual provost correct? True.

---

**CLEVR and CLEAR**
There is a very large metal tourmaline tetrahedron at (-0.44, -1.46), a small glass aquamarine calendar at (-0.51, 1.03), a small amber gray remote control at (-1.67, -1.44), a small amber tourmaline remote control at (-1.32, -0.87), a very large amber apatite printer at (1.45, -1.46), and a small amber labradorite pen at (-1.08, -1.18). Is the number of labradorite amber pens right of the glass calendar greater than the number of tiny tourmaline remote controls that are in front of the tourmaline amber remote control? no. A sikh brother says it's fine, a sister from a different city says it's correct. Is the sister from a different city correct? True. The last object has been removed. Are there more small tourmaline amber things right of the tourmaline remote control than gray things in front of the very large amber printer? no. A sikh brother says yes, a sister from a different city says there's no error. Is the sister from a different city correct? True.

---

### 3.4 FINE-TUNING MODELS

We fine-tune two pre-trained language models to enhance their logical reasoning and misinformation detection capabilities: i) GPT-2 (small), a model with 125M parameters (Radford et al., 2019), fine-tuned using Low-Rank Adaptation of Large Language Models (LoRA) (Hu et al., 2022a), a Parameter-Efficient Fine-Tuning (PEFT) technique; ii) LlaMA 3.2 (small), a model with 1B parameters (Meta Llama Team, 2024), fine-tuned with AdaLoRA (Zhang et al., 2023), which dynamically adjusts parameter importance during fine-tuning. Both models are trained using a language modeling objective (next-step prediction with cross-entropy loss) and early stopping based on validation loss. To address misinformation detection, we introduce an additional binary classification loss, weighted ten times more heavily than the language modeling loss. This binary loss determines whether the model can correctly classify a sentence as truthful or not. For input, we use a prompt like: "Is the previous text truthful? Reply with True or False." We then assess whether the predicted probability for the *True* token exceeds that of the *False* token. To further improve robustness, we add noise to

the embedding vectors using NEFTune (Jain et al., 2024), a technique designed to enhance model performance by introducing controlled perturbations.

## 4 EXPERIMENTAL EVALUATION

In this section, we present the experimental evaluation conducted to assess the effectiveness of the proposed data augmentation approach w.r.t. the considered misinformation detection task. We begin by introducing the datasets used in the evaluation, followed by a discussion of the results obtained.

### 4.1 MISINFORMATION DATASETS

In order to improve the logical abilities of LLMs, we propose augmenting a *Target Task Dataset* (TTD) with data generated using LogicJitter. We evaluate our approach on three datasets used as TTDs: *PubHealth* (Kotonya & Toni, 2020b), *VitaminC* (Schuster et al., 2021) and *ISOT* (Ahmed et al., 2017). *PubHealth* comprises 11,832 claims for fact-checking across a wide range of health-related topics, including biomedical subjects and government healthcare policies. *VitaminC* is a multi-domain fact-verification dataset based on Wikipedia edits, containing 488,904 data points. *ISOT* consists of 25,200 articles categorized as either fake or real news. The truthful articles were obtained by crawling *Reuters.com* while the fake news were collected from websites flagged by *Politifact*. To evaluate the effectiveness of our data augmentation technique, we compare it against two baselines: fine-tuning directly on the TTD and augmenting the TTD using an existing human-labeled misinformation dataset, VitaminC in our case. For consistency, we limit the augmented TTD size to no more than twice the original TTD size. In our pipeline, we first augment the dataset, for example ISOT, by adding an extra amount of data points generated using either LogicJitter or VitaminC. The resulting dataset is then shuffled to ensure randomness before training the model.

Both PubHealth and VitaminC provide claims along with a corresponding piece of evidence that either supports or rejects the claim for each data point. For a data point consisting of evidence $x$, claim $y$, and label $z$, we format the input to the LLM as: 'Evidence: $x$. Claim: $y$. Does the evidence support the claim? Reply with *True* or *False*: $z$'. This structure ensures clarity for the model and directly addresses the task of evaluating the relationship between the claim and its evidence. In contrast, ISOT provides longer claims $x$, with corresponding labels $z$, but without associated evidence. For this dataset, we format the input as: '$x$. Is the preceeding text likely truthful and not fake news? reply with *True* or *False*: $z$'.

### 4.2 RESULTS

Table 2 illustrates the results in terms of *accuracy* of misinformation detection (i.e., classifying claims into true and fake). As it emerges from the table, augmenting the TTD both with LogicJitter and with human-labeled data, such as VitaminC, improves the validation and the test results over the PubHealth dataset. Surprisingly, LogicJitter achieves this without the need for human-labeled data. When the TTD is VitaminC instead, can see that the best performance is achieved with LogicJitter with errors and random characters, but without problem revision.

Table 2: Ablation study to understand which parts of LogicJitter contribute the most. The full LogicJitter is composed by four parts: G stands for game description, E for including generations with errors, C for including random characters, and full stands for GECR, with R for revisions. We compare also to augmenting with the human labelled dataset VitaminC. We show results on the PubHealth and VitaminC datasets using the GPT2 model and fine-tuning with LoRA.

|  | **PubHealth** | | **VitaminC** | |
| --- | --- | --- | --- | --- |
| **augmentation** | val | test | val | test |
| TTD | 54.8% | 54.9% | 55.0% | 55.5% |
| TTD + VitaminC | **69.8**% | **68.2**% | | |
| TTD + LogicJitter (G) | 42.7% | 42.1% | 74.5% | 74.4% |
| TTD + LogicJitter (GE) | 67.4% | 66.2% | 74.9% | 74.8% |
| TTD + LogicJitter (GEC) | 66.5% | 64.8% | **75.0**% | **75.1**% |
| TTD + LogicJitter (full) | 68.1% | 66.2% | 59.9% | 59.7% |

We can also observe in Table 3, that both adding and removing data from the PubHealth with the augmentation data were effective in improving generalization of the classification model in terms of test accuracy. Instead, when the TTD were VitaminC and ISOT, and using both GPT2 and Llama, and both LoRA and AdaLoRA, increasing the amount of data was generally better.

Table 3: Test accuracy for different augmentation strategies on the PubHealth, VitaminC and ISOT datasets using GPT2 125M parameters and LLama3.2 1B parameters. We show how much additional data we add to ($+$) or remove from ($-$) the TTD with augmentation. Almost any amount of data augmentation improves performance. LogicJitter (LJ) achieves it without the need of human annotated data.

| dataset | PubHealth | | VitaminC | ISOT |
| --- | --- | --- | --- | --- |
| model | GPT2 | | GPT2 | Llama |
| PEFT | LoRA | | LoRA | AdaLoRA |
| **augm.** | **+LJ** | **+VitC** | **+LJ** | **+LJ** |
| +100% | 66.2% | 68.2% | 59.7% | **75.0**% |
| +75% | 68.6% | 73.2% | 54.8% | |
| +50% | 62.0% | 66.5% | 65.5% | 71.4% |
| +25% | 50.2% | 65.1% | **93.0**% | 58.1% |
| TTD | 54.9% | | 55.5% | 65.3% |
| -25% | 74.2% | 70.4% | 53.7% | 74.0% |
| -50% | **79.9**% | 69.9% | 50.8% | 51.9% |
| -75% | 78.9% | 74.9% | 63.6% | |
| -100% | 55.9% | **79.1**% | 51.4% | 19.5% |

## 5 DISCUSSION AND CONCLUSIONS

We have introduced LogicJitter, a data augmentation technique for misinformation detection that is generated algorithmically and therefore does not require human labelled data. We showed that it successfully improved the generalization ability of the model compared to fine-tuning only on the target dataset, or augmenting with existing human labeled data on misinformation. It therefore turns an expensive task, expert labeling, into a cheap task, algorithmic generation. Being able to generate a dataset algorithmically comes with a few convenient factors, such as the fact that is completely balanced in terms of number of true and false statements, and stereotyping biases are completely absent since it is coded in such way. Somewhat ironically, we use the often considered old school rule-based AI, such as context-free grammars and causal networks, to compensate for the shortcomings of the new wave of Deep Learning based AI. We believe our evidence supports **RQ1** and **RQ2** in the positive: rule-based games, inspired by the attempt to compensate for cognitive biases and fallacies, can improve LLMs logical reasoning, shown by their improved ability to detect misinformation.

Another possibility to what we presented would be to use existing human datasets such as GLUE (Wang et al., 2018), and present the wrong label to ask the LLM to estimate the veracity of the answer. We decided however to stick to a purely rule-based approach, to cleanly verify its effectiveness. As a possible future direction, it is interesting to consider fuzzy logic statements, to provide the games with more flexibility to deal with uncertainties. It will also be interesting to attempt at compensating for more cognitive biases and fallacies. Moreover, a rule-based approach such as LogicJitter, could be used to produce an algorithmically generated explanation on why an answer is right or wrong, and could therefore be useful for those interested in explainability.

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
