# OpenReview forum: "LogicJitter: Let LLMs play Logic Games and they will Detect Misinformation"
_ICLR.cc/2025/Conference — Submitted to ICLR 2025_

### Official Review · Reviewer_mTBh · 2024-11-01

**Soundness:** 2
**Presentation:** 2
**Contribution:** 2
**Rating:** 3
**Confidence:** 4

**Summary:**

This work proposes LogicJitter, an approach to detect misinformation to augment training data and improve LLM reasoning. Experiments on various datasets demonstrate that this approach outperforms various baselines.

**Strengths:**

+ misinformation detection is an important research topic
+ the connection from logic games to misinformation analysis is interesting

**Weaknesses:**

- the experiments are underwhelming
- the technical contribution is not clear

**Questions:**

- I wonder if the authors have considered any valid baselines to misinformation detection. There are encoder-based LM-based approaches, LLM-based approaches, graph-based approaches, etc. For now the baseline selection seems largely inadequate.

- I wonder if the authors could better highlight the technical contribution of this approach. To me it seems to mostly involve the generation of synthetic data at training time, which is somewhat trivial.

- The design choices of the synthetic data generation could be better validated by ablation studies.

- Overall the paper only has 8 pages of content, leaving much space for extensive experiments and analysis. I hope the authors could strengthen the experiments to better highlight the strength of this approach.

- The font might be different from the template's default, which might be desk rejection likely.

---

> ### Author Response · Authors · 2024-11-27
> **Reply to Reviewer mTBh**
>
> Dear reviewer,
>
> as you and other reviewers suggested, we added more datasets, and more and bigger models. I am giving the same reply to every reviewer since some criticism is shared, and possibly you've read the other reviews, but I can't expect you to read each reply to each review.
>
> We do think it is a proof of concept, and we would like to keep exploring this direction, but we do find the initial results to be very promising, and that's why we think it is fair to share the news with the community. We are also aware that as someone said, the contribution can be perceived as trivial, but it is somewhat the goal, to create a dataset that is ridiculously easy to create, but whose impact on learning might be surprisingly valuable in downstream tasks. Especially given that LLMs are known to be weak in algorithmic tasks where GNNs excel.
>
> We also tried to rewrite the training and dataset descriptions more clearly since we realized we could have done it better. About the template, we were using the 2024 template. Now you can see the line number on the left that appears on the 2025 template. However I don't understand why it is not showing the line 'Under review as a conference paper at ICLR 2025'.
>
> About the fair criticism of including a PyTorch package release as a contribution, we think it is a scientific contribution to make it easy to reproduce, and that's why we state it.
>
> Thanks a lot for the feedback!

---

### Official Review · Reviewer_VVk5 · 2024-11-04

**Soundness:** 2
**Presentation:** 1
**Contribution:** 1
**Rating:** 1
**Confidence:** 4

**Summary:**

This paper proposes LogicJitter, a data augmentation technique that enhances the logical reasoning abilities of LLMs  for improved misinformation detection. The method generates both correct and incorrect statements within rule-based logic games to counteract human cognitive biases, offering a cost-effective alternative to large labeled datasets.

**Strengths:**

1. simple and straight idea

**Weaknesses:**

1. Providing a pytorch and huggingface package cannot be claimed as a research contribution, they are basically the engineering problem.
2. There’s no technical novelty in this paper, authors simply collect some logical tasks and finetune LLM to enhance misinformation detection. Moreover, the mechanism behind the generalization from logical tasks to misinformation detection task is not well studied.
3. The GPT-2 small model has only 124M parameter size and cannot be classified as as an LLM. In LLM with billions of parameters, the model has seen tons of facts and reasoning data, I am concerned that the proposed training data paradigm might not generalize to that large LLMs.
4. The table 3 is not clear, why the left row has a none value, and what does the 47.2\% and 46.8\% mean?

**Questions:**

please see the weaknesses above

---

> ### Author Response · Authors · 2024-11-27
> **Reply to Reviewer VVk5**
>
> Dear reviewer,
>
> as you and other reviewers suggested, we added more datasets, and more and bigger models. I am giving the same reply to every reviewer since some criticism is shared, and possibly you've read the other reviews, but I can't expect you to read each reply to each review.
>
> We do think it is a proof of concept, and we would like to keep exploring this direction, but we do find the initial results to be very promising, and that's why we think it is fair to share the news with the community. We are also aware that as someone said, the contribution can be perceived as trivial, but it is somewhat the goal, to create a dataset that is ridiculously easy to create, but whose impact on learning might be surprisingly valuable in downstream tasks. Especially given that LLMs are known to be weak in algorithmic tasks where GNNs excel.
>
> We also tried to rewrite the training and dataset descriptions more clearly since we realized we could have done it better. About the template, we were using the 2024 template. Now you can see the line number on the left that appears on the 2025 template. However I don't understand why it is not showing the line 'Under review as a conference paper at ICLR 2025'.
>
> About the fair criticism of including a PyTorch package release as a contribution, we think it is a scientific contribution to make it easy to reproduce, and that's why we state it.
>
> Thanks a lot for the feedback!

---

### Official Review · Reviewer_KMPC · 2024-11-04

**Soundness:** 1
**Presentation:** 2
**Contribution:** 3
**Rating:** 3
**Confidence:** 4

**Summary:**

The paper proposes generating synthetic data through logic games, targeted specifically at a large number of cognitive biases / logical fallacies. Then, an LLM is fine-tuned on this data (along with domain data, i.e., misinformation detection data). This improves performance on a misinformation detection dataset. Importantly, the augmenting data here also does not need human labeling, since it is generated from the rule-based logic games, potentially reducing the need for such labeled data that can be expensive and very difficult to obtain in large quantities in this domain.

**Strengths:**

Idea is very cool and could potentially be applied to many domains where reasoning is important. Perhaps even towards mitigating bias and making more fair LLMs.

Covers many common biases/fallacies.

**Weaknesses:**

Mainly, the experimental results presented do not sufficiently prove this works. In particular:

The main evaluation is only done on one dataset, and the paper suggests that the dataset may not be high quality considering fine-tuning on any amount of it is worse than fine-tuning on any of the other datasets considered (Table 3, -100% row) - and by an extremely large margin in the case of VitaminC. This seems odd, and to indicate something might be unusual about this dataset. Why not test with VitaminC as target dataset, considering that one is already implemented? Even more datasets would be better. Testing on one dataset alone would already be cause for hesitation, not to mention when there seems to be something weird about it.

The PubHealth dataset has imbalanced class distribution, so evaluating with accuracy alone could potentially give misleading results (e.g., if some training procedure biases towards one class rather than actually improving reasoning). It seems important to add some other metrics, e.g., macro F1.

What about performance without fine-tuning? That seems an important baseline.

Model (GPT-2) is old and outdated. It seems important to assess if this can still improve modern models. For example, there are Llama-3.2 or Gemma 2 models now with comparable size to GPT-2, not to mention larger ones.

This could be a great idea. But it really needs more empirical evidence and thoroughness to back it up.

**Questions:**

How do you parse answers to get them into a form where they can be compared with the label? Could that be affecting the result in any way?

At end of intro it says "pre-training" but at beginning of methodology it says "fine-tuning", bit confusing. Should probably be fine-tuning throughout.

The style template appears to be incorrect, though it's below page limit by a substantial amount so it seems it would be fine if in the correct template.

---

> ### Author Response · Authors · 2024-11-27
> **Reply to Reviewer KMPC**
>
> Dear reviewer,
>
> as you and other reviewers suggested, we added more datasets, and more and bigger models. I am giving the same reply to every reviewer since some criticism is shared, and possibly you've read the other reviews, but I can't count on you reading each reply to each review.
>
> We do think it is a proof of concept, and we would like to keep exploring this direction, but we do find the initial results to be very promising, and that's why we think it is fair to share the news with the community. We are also aware that as someone said, the contribution can be perceived as trivial, but it is somewhat the goal, to create a dataset that is ridiculously easy to create, but whose impact on learning might be surprisingly valuable in downstream tasks. Especially given that LLMs are known to be weak in algorithmic tasks where GNNs excel.
>
> We also tried to rewrite the training and dataset descriptions more clearly since we realized we could have done it better. About the template, we were using the 2024 template. Now you can see the line number on the left that appears on the 2025 template. However I don't understand why it is not showing the line 'Under review as a conference paper at ICLR 2025'.
>
> About the fair criticism of including a PyTorch package release as a contribution, we think it is a scientific contribution to make it easy to reproduce, and that's why we state it.
>
> Thanks a lot for the feedback!

---

### Official Review · Reviewer_pXJX · 2024-11-16

**Soundness:** 1
**Presentation:** 2
**Contribution:** 1
**Rating:** 3
**Confidence:** 4

**Summary:**

This work proposes LogicJitter,  an approach to enhance the logical reasoning abilities of LLMs for improved misinformation detection. Experiments demonstrate that this approach outperforms baselines.

**Strengths:**

1. The paper is clear and straightforward.

**Weaknesses:**

1. The technical contribution is limited
2. The experiments are far from adequate
3. The paper does not use an ICLR template and probably should be desk rejected.

**Questions:**

see weakness

---

> ### Author Response · Authors · 2024-11-27
> **Reply to Reviewer pXJX**
>
> Dear reviewer,
>
> as you and other reviewers suggested, we added more datasets, and more and bigger models. We do think it is a proof of concept, and we would like to keep exploring this direction, but we do find the initial results to be very promising, and that's why we think it is fair to share the news with the community. We are also aware that as you said, the contribution can be perceived as trivial, but it is somewhat the goal, to create a dataset that is ridiculously easy to create, but whose impact on learning might be surprisingly valuable in downstream tasks. Especially given that LLMs are known to be weak in algorithmic tasks where GNNs excel.
>
> We also tried to rewrite the training and dataset descriptions more clearly since we realized we could have done it better. About the template, we were using the 2024 template. Now you can see the line number on the left that appears on the 2025 template. However I don't understand why it is not showing the line 'Under review as a conference paper at ICLR 2025'.
>
> About the fair criticism of including a PyTorch package release as a contribution, we think it is a scientific contribution to make it easy to reproduce, and that's why we state it.
>
> Thanks a lot for the feedback!

---

### Meta-Review · Area_Chair_HT1w · 2024-12-19

**Metareview:**

**Summary:**

The authors propose using synthetic logic game data augmentation to enhance LLM reasoning for misinformation detection. They finetune a baseline model on the synthetic data, and show improvement on fact-checking/misinformation datasets.

**Strengths:**

- Promising concept for improving LLM misinformation detection abilities that could be extensible to other tasks

**Weaknesses:**

- Unclear technical contribution

- Inadequate experimentation and evaluation

- Lack of F1 results (the paper only focuses on accuracy and ignores dataset imbalance)

This is a clear rejection, but I encourage the authors to refine the work based on all the reviewers' feedback and try for another venue in the future.

**Additional Comments On Reviewer Discussion:**

The reviewers are unanimously against acceptance, and I agree that the paper is not at a publishable stage. There are several serious technical errors pointed out by the reviewers and highlighted above. These have not been adequately addressed by the authors in the rebuttal, and would require a complete revision of the paper to address, after which it is unclear if the paper's conclusions would remain consistent. The authors' response is generic to all reviewers, does not answer reviewers' individual questions and does not argue for the paper's technical contribution to the research community (which seems minimal) or better motivate the connection between logical games and misinformation detection.

---

### Decision · Program_Chairs · 2025-01-22

Reject